# Mechanical Properties of Cast-in Anchor Bolts Manufactured of Reinforcing Tempcore Steel

**DOI:** 10.3390/ma12132075

**Published:** 2019-06-27

**Authors:** František Bahleda, Petra Bujňáková, Peter Koteš, Lívia Hasajová, František Nový

**Affiliations:** 1Laboratory of Civil Engineering, University of Žilina, Univerzitná 8215/1, 010 26 Žilina, Slovakia; 2Department of Structures and Bridges, Faculty of Civil Engineering, University of Žilina, Univerzitná 8215/1, 010 26 Žilina, Slovakia; 3Dubnica Institute of Technology, DTI University, Sládkovičová 533/20, 018 40 Dubnica nad Váhom, Slovakia; 4Department of Materials Engineering, Faculty of Mechanical Engineering, University of Žilina, Univerzitná 8215/1, 010 26 Žilina, Slovakia

**Keywords:** reinforcing steel bars, yield strength, ultimate strength, bolts, Tempcore rebars

## Abstract

The tempcore process is implemented in rolling mills to produce high strength reinforcing steel. Besides being used as reinforcement, rebars are also used as the base material for the manufacturing of anchor bolts. The mechanical properties of reinforcement bars used in Europe are assessed in accordance with Eurocode without the recommendations for cast-in anchor bolts. The material properties of Tempcore rebars are not homogenous over the bar cross section. The European Assessment Document (EAD) for the cast-in anchor bolts does not exactly specify the mechanical properties of the thread part. The aim of these experiments is to show the different mechanical properties of rebars and their thread parts. The experiments were performed on rebars modified by peeling to characterize the reduction of diameter in a thread part. As a possible way to predict mechanical properties in a non-destructive way, the hardness tests were performed. Next, the application of the correlation relationship between hardness and tensile strength has been determined. The paper formulates preliminary recommendations for assessment of the cast-in anchor bolts in practice.

## 1. Introduction

The mechanical properties of reinforcement bars used in Europe are assessed in accordance with Table C.1 of EN 1992-1-1 [1] by characteristic values of yield strength, a minimum ratio f_t_/f_y_, and a minimum characteristic strain at maximum force ε_uk_. Those characteristics are determined by destructive testing. As the ultimate limit state of reinforced concrete structures is most often characterized by development of plastic hinges, the three earlier mentioned characteristics are assumed to be enough to guarantee a ductile behaviour of the structure at the ultimate limit state (UTS). Besides being used as reinforcement, rebars are also used as the base material for the manufacturing of anchor bolts (see Table 3.1 of EN 1993-1-8 [2]). Such bolts are typically manufactured by peeling off the ribbed perimeter of the bar and cutting or rolling a thread on the bar. They are used to anchor concrete or steel connection for concrete structures (e.g., foundations, base columns, slabs, walls, and similar applications). Typically, the anchor bolts are either headed or straight. Headed bolts are used mainly in shallow structures for end anchoring, whereas the straight bolts are used for lap splices [3,4,5]. Each cast-in anchor bolt includes one nut and washer or two nuts and washers depending on the application, see Figure 1. One end is intended to be cast in concrete, while the opposite end is threaded and projects from the concrete. They are usually cast into reinforced concrete and transfer loads. In the absence of a European standard for anchor bolts, including the material recommendation, the structural performance of such bolts is assessed by destructive testing in accordance with the European Assessment Document (EAD) [6].

The characteristic value of tensile resistance of anchor bolts under the static and quasi-static actions is determined in accordance with [6] as follows: (1)$$N_{Rk,s,calc}=A_{s}f_{uk}$$
where *A_s_* is the stress area of the thread and *f_uk_* refers to the characteristic tensile strength of the bolt. 

## 2. Material Properties

The cast-in anchor bolts are mostly fabricated of ribbed bars with specifications according to Table 1. The majority of rebars available on the European market are manufactured by the Tempcore process. The Tempcore method was developed by the research centrum “Centre de Recherches Métallurgique” (CRM) in 1974 in Belgium [7]. This process increases the yield and ultimate tensile strength, ductility, and bendability of reinforcing bars. The process is divided into the three stages: (I) quenching of the surface layer, (II) self-tempering of the martensite, and (III) transformation of the core. The first stage consists of rapid cooling for a short time after the rebar leaving the last rolling stand. The surface layer of the bar is quenched into martensite and the core remains austenitic. The next stage is the tempering of the martensite layer, the heat releases from the core to the surface. The last stage is the transformation of the core from austenite into ferrite and perlite or into bainite, ferrite, and perlite. Therefore, three layers with different microstructural features (surface layer, transition layer, and core) can be observed in the cross section of the rebar. These three different layers have different mechanical properties. The ferrite is very ductile but soft and martensite is very hard but very brittle. Values of yield and ultimate strength in the outer layer are higher and they decrease gradually in the core [7,8,9]. The final microstructure depends on the chemical composition, bar diameter, rolling end temperature, and cooling intensity in the first stage [7,9,10,11]. The study [9] shows the changes in temperature within the reinforcing bar, which was cooled with the same water flow after reheating. According to this analysis, the temperature of 900 °C is recommended in the reheating process for the achievement of balanced mechanical properties. To obtain a homogenous Tempcore treatment, the intensity of cooling must be high enough to obtain a complete and regular martensite outer ring [12]. 

The evaluation of mechanical properties of reinforcing bars is essential, especially in reinforced structures where inadequate design procedure may cause the risk of premature failure [13,14,15]. The non-standard bars manufactured with poor quality control have a yield strength often lower than minimum specific value [16,17]. The poor performance has serious aspects, especially for seismic applications. Research studies [18,19,20,21,22,23] have shown the effect of the strain rate on the tensile properties of the rebars with yield stresses ranging from 290 to 710 MPa. The ratio between ultimate tensile strength and yield stress decreases as the strain rate increases. The strain rate sensitivity decreases from the inner to the outer layers. The lower strength steel is more susceptible to strain rate effects compared to the higher strength steel. 

## 3. Experimental Program

The European Assessment Document [6] for the cast-in anchor bolts does not exactly specify whether the characteristic yield strength (f_yk_ > 500 MPa) should cover a nominal diameter of reinforcing bar or a thread diameter. Therefore, the experimental program has been focused on verification of mechanical properties depending on the shape of the reinforcing bars. 

### 3.1. Experimental Procedure

In order to achieve a better understanding of mechanical properties of the individual layers, tensile tests and hardness tests were performed on the Tempcore rebars made of B500B. The experimental testing was divided into three steps:

**Step 1:** Tensile test of rebars produced by manufacturer A: ribbed steel bars (Tempcore bars) with three different diameters (25, 16, and 10 mm) were tested. The following series were tested for each diameter:Unmodified rebarRebar with 1/6φ removed by peelingRebar with 1/3φ removed by peelingRebar with 2/3φ removed by peeling

**Step 2:** Tensile tests of 25 mm rebars produced by several manufacturers A, B, C, D, where mechanical properties of the supplementary layers of 1/24φ, 2/24φ, 3/24φ, 1/2φ, 3/4φ, and 4/5φ removed by peeling were verified.

**Step 3:** Vickers hardness tests of 25 mm rebars produced by several manufacturers A, B, C, D and rebars of a 16 mm (φ16A) and 10 mm (φ10A).

### 3.2. Tensile Test

Tensile tests were performed in accordance with standards EN ISO 6892-1 and EN ISO 15630-1 [24,25] using the tensile testing apparatus shown in Figure 2. Three identical samples of each specimen were tested. The original gauge length L_o_ of each sample followed the standard [24] was expressed as Lo = k √So, where k is a coefficient of proporcionality (k = 5.65). When the cross-sectional area of the test specimen is too small the higher value k = 11.3 is preferable. The original cross-sectional area S_o_ is the average cross-sectional area calculated from the measurements. The force was applied as axially as possible to minimize bending and did not exceed a value corresponding to 5% of the specified yield strength. 

### 3.3. Hardness Test

The Vickers method of an identation hardness testing was chosen for determining the hardness of rebars and the assessment of correlation relationship between hardness and tensile strength. The hardness test was executed with an automatic machine Zwick/Roel ZHVμ-A according to EN ISO 6507-1 [26] on rebars in Figure 3. The measurements were performed in air at room temperature using the load of 500 gf for a holding time of 10 s. The largest and smallest values were discarded and then the average of the remaining values was obtained for evaluation.

## 4. Experimental Results and Discussion

### 4.1. Microstructures of Tested Rebars

The microstructure of the rebars was revealed for better understanding of the relationship between strength and hardness using a microscope Zeiss Axio Imager A1 (Jena, Germany). Each specimen was prepared using a standard metallographic procedure to minimize the damage in the microstructural preparation stage. The specimens were ground and polished with 1 µm diamond paste using Tegramin-30 (Struers) machine and etched with 2% Nital (2% HNO_3_ in ethanol). Figure 4 shows the microstructure of the cross section of rebar with a diameter of 25 mm. Near the surface, the microstructure consists of fine-grained tempered martensite. A mixture of ferrite and pearlite is in the core. The ferritic-pearlitic microstructure of the core is relatively coarse-grained and pearlite is present in both lamellar and partially decayed in globular form. Ferrite occurs also in the form of the Widmanstätten pattern resulting from the formation of a new phase along certain crystallographic planes of the parent solid solutions (austenite) in the orientation of the lattice in the parent phase. The Widmanstätten ferrite plates emanate from prior austenite grain boundaries into the remaining pearlitic-ferritic matrix.

### 4.2. Assessment of Tensile Tests

The comparison of the mechanical properties over the cross section of the reinforcing bars with different diameters (φ25, φ16, φ10) is reported in Table 2. During the test, the yield strength f_y_ and tensile strength f_t_ were measured. The characteristic yield strength f_yk_ (YS) and the tensile strength f_tk_ (TS) were determined from three samples considering the 5%-fractile of the failure loads measured in the test. The f_y,min_ is the minimum value of the YS in the test series. The f_t,min_ is the minimum value of the TS in the test series.

The cross section area of the unmodified rebar was expressed in two forms, the nominal cross section area (A_s_) and the statically effective area (A_s,t_). The nominal cross section area is the area of the reinforcing steel bar with taking the ribs into account. The statically effective area was applied only for the cross section of the steel bar without the ribs. The statically effective cross section area A_s,t _was determined from the mass of the test piece, the length (one meter long), and from its density. Resulting in the calculation of the strength, the statically effective area may be used for better interpretation of the stress distribution. 

The results from the tensile tests were used for determination of mechanical properties of individual layers (yield strength f_y,i_ and tensile strength f_t,i_) shown in Table 2. The average mechanical properties of the reinforcement bars with diameter 25 mm (φ25A), through individual layers depending on the reinforcement radius are shown in Figure 5. The reinforcing steel achieves the YS in the core of 385 MPa and in the layer near the surface the YS ranges from 735 to 795 MPa. The TS measured in the core was approximately 524 MPa and in the layer near the surface the TS ranges from 794 to 809 MPa.

Figure 6 shows the measured values of YS and TS of the samples with basic diameter 25 mm (φ 25, producer A) depending on the modified diameter shape. The measured value of YS and TS has higher scatter in the core. The lowest values of YS and TS were observed at the core and the higher value at the surface layer. The bold numbers in Table 2 indicate the diameters of the core areas of threads M10, M16, and M24. The measurements show that with all the three diameters the TS at these diameters is higher than 550 MPa (value governing the tensile strength of the bolt in accordance with Equation (1)). The YS is lower than 500 MPa only in specimen φ25 A, Figure 7. 

Figure 8 illustrates the strain-hardening potential, the ratio of TS to YS (f_t_/f_y_). The ratio ranges between 1.1 (r/r_t_ = 1.0) and 1.23 (r/r_t_ = 0.35) for bars with diameter of 10 mm. The strain hardening for bars with diameter of 16 mm ranges from 1.17 (r/r_t_ = 1.0) to 1.34 (r/r_t_ = 0.34) and for bars with a diameter of 25 mm it ranges between 1.17 (r/r_t_ = 1.0) and 1.36 (r/r_t_ = 0.21). It is observed that strain hardening starts to increase in the modified shape of the rebar by peeling (high strain-hardening potential) compared to unmodified rebars. The strength is higher for specimens with a low value of TS/YS.

Similar conclusions can be made by evaluating the measurements done on samples B, C, D. Therefore, the next analysis was focused on the Tempcore bars with diameter of 25 mm intended to use for anchor bolts manufactured by several producer (A, B, C, D).

A nominal cross-sectional area (A_s_) was considered in calculation of YS and TS of the whole ribbed rebars (f_y_, f_t_). The mechanical properties of the core were specified on the modified rebar with 1/3 of diameter (1/3d). The measured cross-sectional area was used by determination of the yield strength of the core f_y,c_ and ultimate strength of the core f_t,c_. Table 3 shows the comparison of mechanical properties of the ribbed bars and the core of the bars.

The average YS of the core (f_y,c_) reaches about 72–80% of the average YS (f_y_) of the rebar with diameter 25 mm. The average TS in the core ranges between 81 and 91% of the average TS (f_t_). The ratio TS/YS, indicates the ductility capacity of the bar. The higher ratio is better for a structure to avoid failure. The actual TS/YS in the core (f_t,c_/f_y,c_) for rebars of 25 mm and 16 mm is more than recommended value of 1.25 [16]. It is observed that the TS of the core f_t,c_ is very similar to YS of the nominal cross section of the rebars with diameters of 25 mm and 16 mm. Further, this assumption should be used for estimating strength of a core, or determination of YS of the reinforcing bar. For a conservative approach, it would be possible to consider ultimate tensile strength f_uk_ equals to f_t,c_ ≈ f_yk_ according to the relation (1) of this work. More extensive research is needed to confirm this hypothesis.

### 4.3. Assessment of Hardness Test

Figure 9 shows the hardness profiles of specimens with diameter of 25 mm manufactured by several producers and typical hardness profiles of specimen of 25, 16, 10 mm (producer A). The hardness of rebars with diameters of 25 mm has a value of 155 HV in the core and maximum 301 HV in the surface layer. From the results observation and comparison, it is clear that the Tempcore rebar consists of three layers (soft core, transition layer, and hard surface layer). The surface layer is about 50% harder than the core of rebars with diameter of 25 mm.

Several hardness conversion formulas had been published to estimate the yield strength f_y_ [27,28,29,30,31,32,33]. Other authors supposed the proportional relationship between ultimate tensile strength f_u_ and the Vickers hardness number HV for materials with approximately the same modulus of elasticity [29,30,31]. A reasonable prediction of UTS (f_u_) may be obtained using the relation:f_u_ = k HV(2)
where k is a proportional characteristic constant, and HV is hardness. The coefficient k is dependent on the type of metal [8]. For many types of steel, the coefficient k is about 3.0 [28,31].

According to [30,32] the ratio of hardness to UTS is lower than 3 in the materials with good ductility. Generally, according to [28], Equation (3) can be used to determine the UTS for quenching steel and Equation (4) for annealed steels:f_u_ (I) = 3.2 × HV − 19.923(3)
f_u_ (II) = 3.6655 × HV – 42.527(4)
where HV is the Vickers Hardness number.

Based on the tensile and yield strength considering the layers measured on specimens with 25 mm diameter and hardness test (Figure 9a), the formulas for yield and tensile strength were expressed by regression analysis. The least-squares linear regression gives the correlation for TS according the Equation (5), where the correlation coefficient attains the value R^2^ = 0.9927:f_u_ (III) = 2.4389 × HV + 131.75 (5)

A least-squares linear regression gives the correlation for yield strength of rebar (25 mm) according to Equation (6):f_y_ (IV) = 3.4803 × HV − 190.59 (6)

The correlation coefficient is R^2^ = 0.9816. 

Figure 10 shows the relationship between TS and HV according to Equations (3) and (4) considering the measured value on rebars φ25. The curves (III) and (IV) represent a linear approximation between hardness and tensile strength (yield strength). The tempcore steel with 25 mm diameter achieves higher strength at a lower hardness level and lower strength value at higher hardness level compared to the conversation Equation (3).

Figure 11 shows the TS and YS as a function of hardness according to Equations (3)–(6) from the core to the surface of the φ25 mm rebar. The f_u,i _represents the TS of the individual layers of the cross section and f_y,i _shows the YS of the individual layers from tensile test (Figure 5). It is noted that in the core the strength is practically constant. 

Finally, Table 4 presents the comparison of YS and TS obtained from the tensile test and the anticipated values calculated from measured value of hardness f_u_(I–IV). The approximation curve of the n-th degree was applied at specific points of the obtained strength. The linear and polynomial functions were used to approximate the function for each series. The Equation (5) reflect the better dependence between the HV and TS using a polynomial function. On the other hand, a linear function has proved better compliance for YS defined according to Equation (6).

The average yield strength of the thread core was determined by integrating YS from hardness. The yield strength was calculated for the maximum (d_3,max_) and minimum (d_3,min_) diameters of the thread. Table 5 shows the comparison of YS between bars with diameters of 10 mm, 16 mm, and 25 mm and different producers (25 A, B, C, D). The yield strength values in thread cores of rebars with 25 mm diameter are lower than requirements in EAD [6].

## 5. Conclusions

The paper presents a verification of mechanical properties of Tempcore rebars modified by peeling. The goal is to characterize the effect of the reduction of rebar diameter on the mechanical properties of cast-in anchor bolts. Based on the experimental results and data presented in the above sections, the following conclusions can be drawn:(1)The tensile strength of all tested rebars modified by peeling is higher than 550 MPa. It is thus appropriate to calculate the characteristic value of resistance of the thread using the characteristic value of tensile strength of the base material B500B (f_uk_ = 550 MPa).(2)The yield strength of tested rebars with diameter 25 mm modified by peeling is slightly lower than 500 MPa. The reduction of strength after peeling is probably related to the microstructure of the rebar. Such reduction does not penalize the structural performance of the anchor bolt, as the tensile capacity of the thread is derived from the tensile strength and not the yield strength of the material of the thread.(3)The experiment shows that the Vickers hardness test is an appropriate method for the prediction of mechanical properties of reinforcement bars, but with less accuracy than the tensile tests. According to experiment, it is possible to derive an empirical relationship between HV and TS with an accuracy of ±12%.(4)In practice, Tempcore rebars are suitable for the manufacturing of cast-in anchor bolts. Of course, the mechanical properties of such anchor bolts have to be confirmed by a continuous quality control performed by the manufacturers.

## Figures and Tables

**Figure 1 materials-12-02075-f001:**
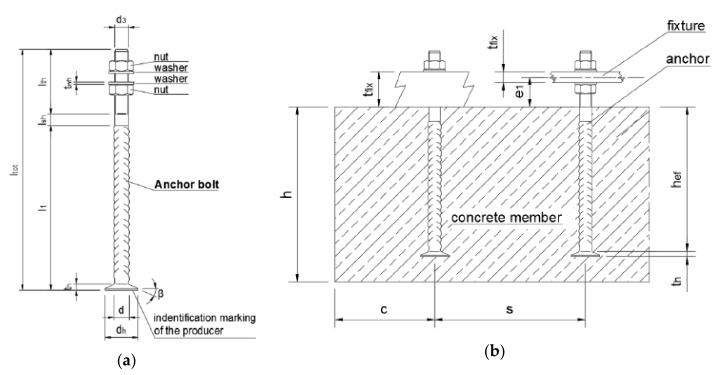
Anchor system: (**a**) Anchor bolt and (**b**) use of anchor bolt.

**Figure 2 materials-12-02075-f002:**
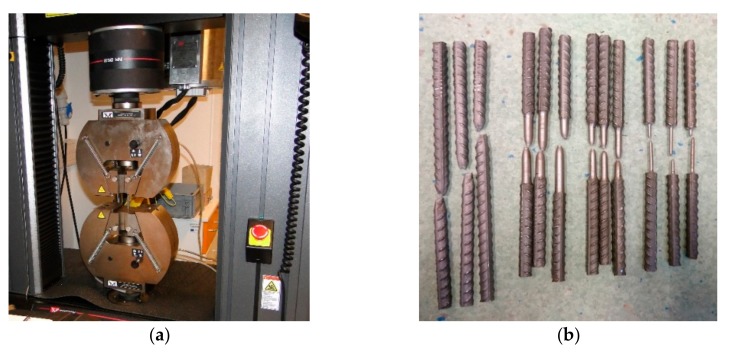
Test procedure: (**a**) Testing machine and (**b**) rebars.

**Figure 3 materials-12-02075-f003:**
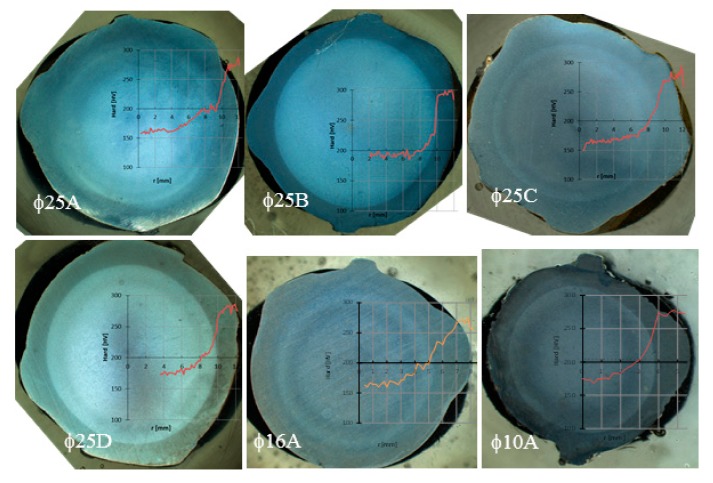
Hardness measurements of different rebars.

**Figure 4 materials-12-02075-f004:**
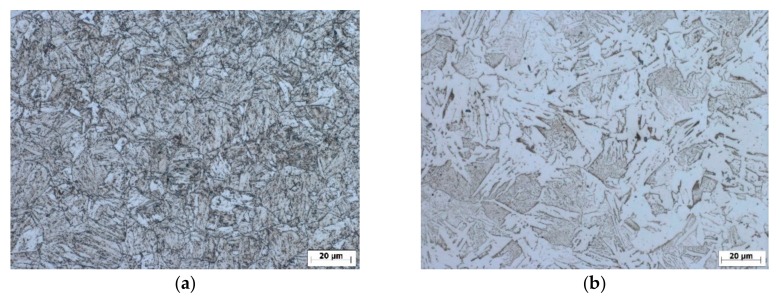
Microstructure over the cross section of the Tempcore bar φ25 (**a**) surface and (**b**) core.

**Figure 5 materials-12-02075-f005:**
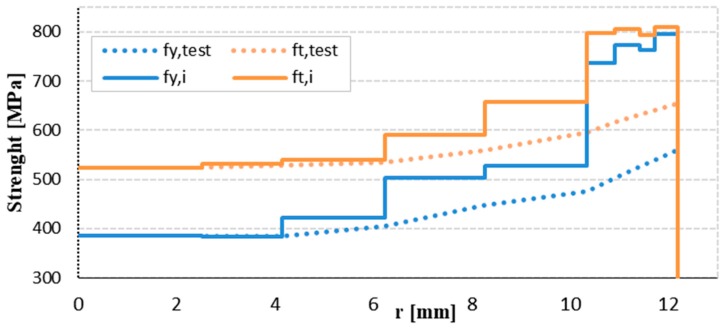
The yield and tensile strength considering the layers (φ25A).

**Figure 6 materials-12-02075-f006:**
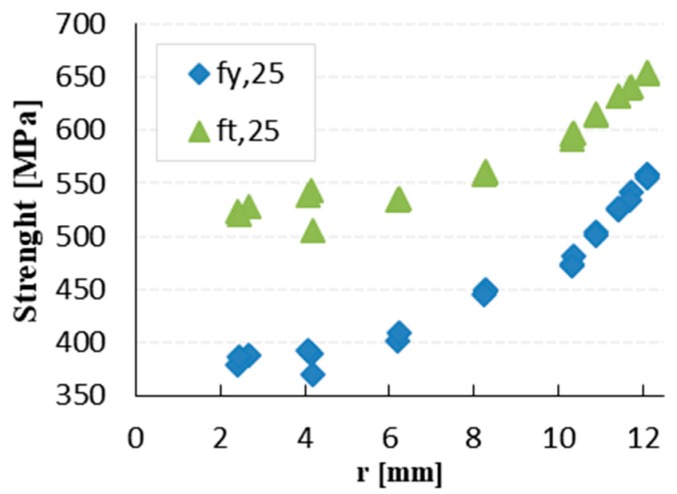
The measured yield and tensile strength of the rebars φ25A.

**Figure 7 materials-12-02075-f007:**
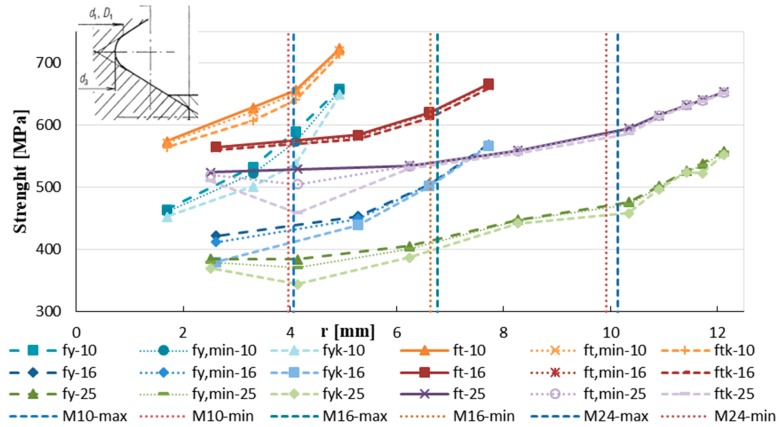
The yield and tensile strengths of the steel bars φ25A, φ16A, φ10A.

**Figure 8 materials-12-02075-f008:**
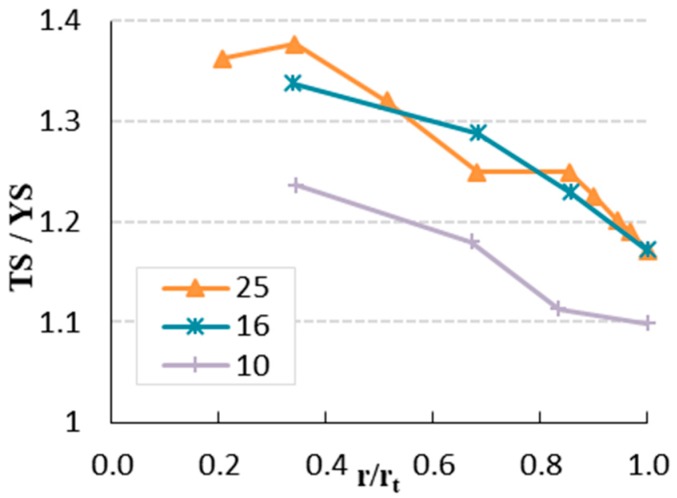
Strain hardening of the rebars.

**Figure 9 materials-12-02075-f009:**
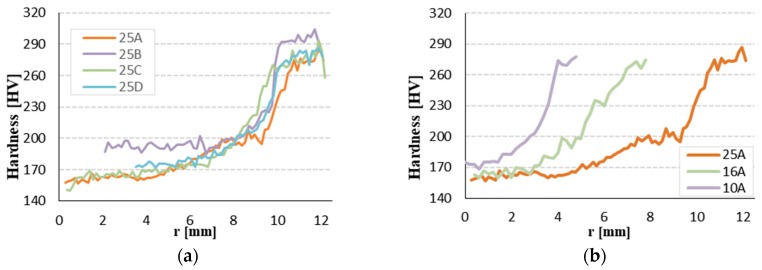
Hardness measurements: (**a**) different producer φ25 mm and (**b**) φ10, 16, 25 mm rebar.

**Figure 10 materials-12-02075-f010:**
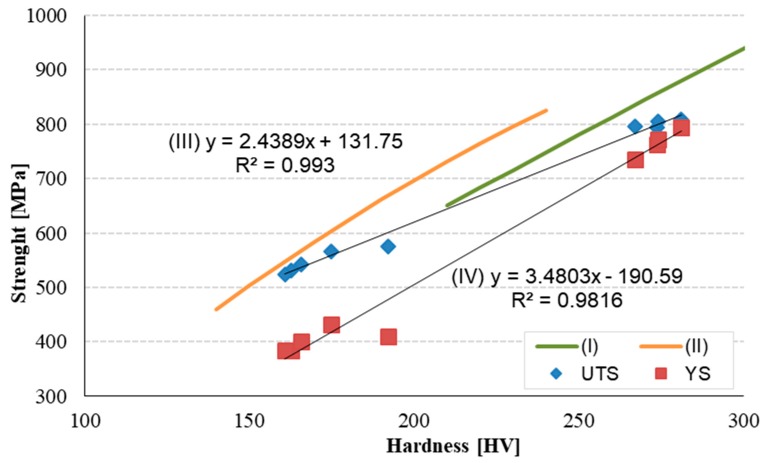
Mechanical properties as a function of hardness (HV).

**Figure 11 materials-12-02075-f011:**
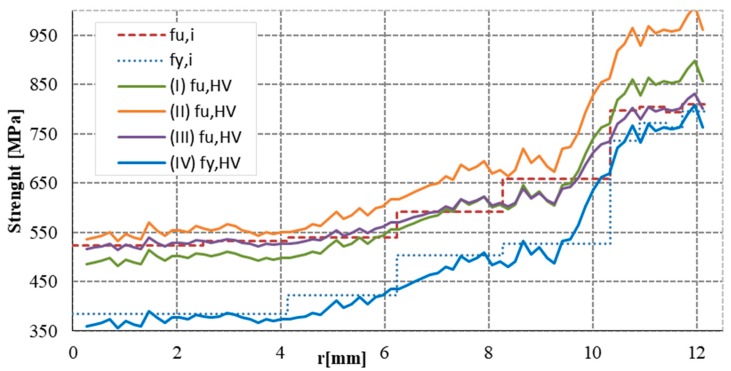
Tensile and yield strength as a correlation from hardness test across the bar (φ25 mm).

**Table 1 materials-12-02075-t001:** Specification for the cast-in anchor bolts [6].

Material Properties/Type of Reinforcing Steel	B500B	B500C
Yield strength (f_yk_)	≥500 N/mm^2^	≥500 N/mm^2^
Ratio of tensile strength over yield strength (f_u_/f_yk_)	≥1.08	≥1.15 ≤ 1.35
Characteristic elongation at maximum force	≥5%	≥7.5%

**Table 2 materials-12-02075-t002:** Mechanical properties of tensile tests.

type	d	A_s_	f_y_	f_t_	f_y,min_	f_yk_	f_t,min_	f_tk_	f_y_/f_y,t_	f_t_/f_t,t_	f_yk_/f_yk,t_	f_yk_/500	f_t_/f_y_
	(mm)	(mm^2^)	(MPa)	(MPa)	(MPa)	(MPa)	(MPa)	(MPa)	(-)	(-)	(-)	(-)	(-)
φ25A	25 *	490.9	524	614	522	519	613	611	0.940	0.940	0.940	1.039	1.171
	24.24 ^x^	461.4	558	653	556	553	653	650	1.000	1.000	1.000	1.105	1.171
	23.44	431.5	538	640	535	522	640	639	0.965	0.981	0.945	1.044	1.191
	22.85	409.9	526	632	526	524	632	630	0.944	0.968	0.948	1.048	1.201
	21.81	373.6	502	615	501	497	615	613	0.901	0.942	0.899	0.993	1.225
	**20.69**	336.1	476	595	472	458	592	586	0.854	0.911	0.830	0.917	1.250
	16.53	214.7	448	559	446	442	558	554	0.803	0.856	0.799	0.884	1.250
	12.46	121.9	405	535	401	387	534	530	0.727	0.819	0.700	0.774	1.320
	8.27	53.8	384	529	370	343	505	459	0.689	0.810	0.622	0.687	1.377
	5.02	19.8	385	524	380	369	520	511	0.690	0.802	0.668	0.739	1.363
φ16A	16 *	201.1	528	619	527	526	617	613	0.929	0.929	0.929	1.052	1.172
	15.42^ x^	186.8	568	666	568	566	664	659	1.000	1.000	1.000	1.132	1.172
	**13.22**	137.2	505	621	504	502	619	611	0.888	0.932	0.887	1.004	1.230
	10.53	87.1	453	584	449	439	582	577	0.797	0.877	0.775	0.878	1.289
	5.23	21.5	422	565	412	379	564	561	0.743	0.848	0.671	0.759	1.338
φ10A	10 *	78.5	639	702	636	630	699	693	0.971	0.971	0.971	1.260	1.098
	9.85^ x^	76.3	659	723	655	649	720	714	1.000	1.000	1.000	1.298	1.098
	**8.22**	53.0	590	656	573	537	652	641	0.895	0.907	0.829	1.075	1.113
	6.62	34.4	533	628	522	501	621	607	0.809	0.868	0.772	1.002	1.180
	3.39	9.0	464	574	461	453	571	564	0.705	0.793	0.697	0.905	1.237

* cross section of the reinforcing steel bar with considering the ribs; ^x^ cross section of the reinforcing steel bar without the ribs; t index shows the properties of the steel bar without peeling (total).

**Table 3 materials-12-02075-t003:** Mechanical properties of the ribbed bars and the core.

Producer	d	d_c_	f_y,c_	f_y,c,min_	f_yk,c_	f_t,c_	f_y_	f_y,min_	f_yk_	f_t_	f_y,c_/f_y_	f_t,c_/ f_t_	f_y_/f_t,c_	f_t,c_/f_y,c_
	(mm)	(mm)	(MPa)	(MPa)	(MPa)	(MPa)	(MPa)	(MPa)	(MPa)	(MPa)	(-)	(-)	(-)	(-)
A	25	8.27	384	370	344	529	524	522	519	614	0.733	0.862	0.991	1.377
B	25	8.24	410	406	396	576	542	541	541	655	0.757	0.879	0.941	1.404
C	25	8.20	401	399	394	543	542	541	540	640	0.741	0.848	0.998	1.354
D	25	8.25	432	414	376	566	542	541	540	644	0.798	0.880	0.957	1.310
A	16	5.23	422	412	380	565	528	527	526	619	0.799	0.912	0.935	1.338
A	10	3.39	464	461	501	574	639	636	630	702	0.726	0.817	1.114	1.237

**Table 4 materials-12-02075-t004:** Comparison of mechanical properties as a function of hardness and tensile testing.

Type	d	YS/f_y,test_	TS/f_u,test_	n degree	f_u_(I)	f_u_(II)	f_u_(III)	f_y_(IV)	p(I)	p(II)	p(III)	p(IV)
	(mm)	(MPa)	(MPa)		(MPa)	(MPa)	(MPa)	(MPa)	%	%	%	%
φ25A	24.24	558	653	6	639	740	652	591	−2.2	13.4	−0.2	6.0
				1	671	757	658	561	2.7	15.9	0.8	0.6
	23.44	538	640	6	630	726	641	569	−1.7	13.3	0.1	5.7
	22.85	526	632	6	621	714	632	552	−1.8	12.9	0.1	4.9
	21.81	502	615	6	605	692	617	523	−1.7	12.5	0.3	4.2
	20.69	476	595	6	588	670	602	495	−1.2	12.6	1.1	4.1
	16.53	448	559	6	547	619	566	433	−2.1	10.6	1.2	−3.3
	12.46	405	535	6	526	592	548	404	−1.7	10.7	2.4	−0.3
	8.27	384	529	6	506	570	533	382	−4.3	7.7	0.7	−0.7
	5.02	385	524	6	504	568	531	380	−3.7	8.4	1.4	−1.3
φ25B	24	588	711	6	731	799	715	652	2.9	12.4	0.5	10.9
				1	716	801	693	610	0.7	12.6	−2.5	3.7
	8.24	410	576	6	596	663	602	480	3.6	15.3	4.5	17.0
φ25C	24.22	577	682	5	690	768	673	582	1.1	12.6	−1.3	0.9
				1	688	769	672	580	0.9	12.7	−1.5	0.4
	8.2	401	543	5	513	567	538	389	−5.6	4.5	−1.0	−3.1
φ25D	24.12	582	691	6	694	774	677	557	0.4	11.9	−2.1	−4.2
				1	674	753	661	564	−2.4	8.9	−4.3	−3.0
	8.24	432	566	6	539	597	557	417	−4.9	5.5	−1.5	−3.6
φ16A	15.42	568	666	5	681	760	666	572	2.3	14.1	0.0	0.6
				1	682	762	667	573	2.4	14.3	0.2	0.8
φ10A	9.85	659	723	5	716	801	697	607	−1.0	10.7	−3.6	−7.9
				1	716	800	697	605	−1.1	10.6	−3.7	−8.1

**Table 5 materials-12-02075-t005:** The yield strength of thread core.

Rebar	Thread	d_,max_	d_,min_	f_y,dmax_	f_y,dmin_	f_yk_ > 500 MPa
		(mm)	(mm)	(MPa)	(MPa)	
φ10	M10	8.128	7.938	533.1	524.9	ok
φ16	M16	13.508	13.271	521.1	514.7	ok
A φ25	M24	20.271	19.840	486.4	477.7	x
B φ25	M24	20.271	19.840	540.5	530.0	ok
C φ25	M24	20.271	19.840	494.4	484.5	x
D φ25	M24	20.271	19.840	483.7	475.9	x

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
