# Peer review of "Mechanical Properties of Cast-in Anchor Bolts Manufactured of Reinforcing Tempcore Steel"

_materials, 2019, doi:10.3390/ma12132075_

Reviewer 1 Report

The manuscript presents an experimental work aimed at mechanical properties of cast - in anchor bolts manufactured of reinforcing steel B500B. I would suggest the following suggestions:

1.    Abstract need to be rewritten to report about the main and new findings obtained in this paper briefly.

2.    The state‐of-art review performed in the introduction section should be strengthened. Special emphasis should be placed on the relationship between microstructure and mechanical behavior of steel.

3.    Please describe the relationship between the size and mechanical behavior, especially in Figure 10.

4.    Please explain the similarities and differences between the results and other researchers.

5. The authors should clarify the aim and the novelty of the study.

Author Response

-              Abstract, Introduction and Conclusions are rewritten. The table 5 is added for better explanation of conclusions.

-              Descriprion of Figure 10 is completed

-              The aim of the experiments is to show the different mechanical properties of rebars and their thread part for cast -in anchor bolts. These bolts are used to anchor  foundations, base columns, slab, e.g. The mechanical properties of reinforcement bars used in Europe are assessed in accordance with Eurocode, but for cast-in anchor bolts there exist only the European Assessment document EAD, which does not specified exactly the tested part or rebars. The study shown the difference between mechanical properties of bars and thread part. Following recommendations for manufacturer of cast-in anchor bolts. (see Conclusions)

Reviewer 2 Report

The manuscript presents the results of research on the basic mechanical parameters of B500B steel, i.e. tensile tests and hardness tests (Vickers hardness tests). The aim of the tests is to determine the values of these parameters in the individual layers of the bar cross-section (from the core to the surface layer). This is very important from the point of view of using B500B steel as the material for the manufacturing of anchor bolts. The authors also suggest how to use the hardness tests (as a non-destructive method) to assess the mechanical properties of the rod. The conclusions drawn are very constructive and useful. Only a few things require minor corrections:

1. The chapter "Experimental results and discussion" is incorrectly numbered (there is 4. but should be 3.; and also 3.1, 3.2)

2. Table 2 - the values of tensile strength (ft) are shown twice; one column is unnecessary

3. Table 4 - one column is not described

4. The last sentence in the conclusion number 2 does not follow directly from the point "Assessment of hardness test" - the names of the methods should be included in the content of this chapter.

Author Response

-              Abstract, Introduction and Conclusions are rewritten. The table 5 is added for better explanation of conclusions.

The recommendations 1,2,3, 4 are corrected in paper.

Reviewer 3 Report

The results and the conlcusions are not sufficiently significant, original, and interesting to warrant publication.

Moreover, the presentation is confusing, since the objective of the research work is not clearly presented and defined, and above all the methodology is not sufficiently well explained that someone else knowledgeable about the field could repeat the study.

Ultimately, the manuscript does not present a specific, easily identifiable advance in knowledge. The results are not applicable and useful to academia or to the profession.

Therefore, I recommend rejecting the submitted manuscript.

Author Response

-              Abstract, Introduction and Conclusions are rewritten. The table 5 is added for better explanation of conclusions.

The aim of the experiments is to show the different mechanical properties of rebars and their thread part for cast -in anchor bolts. These bolts are used to anchor  foundations, base columns, slab, e.g. The mechanical properties of reinforcement bars used in Europe are assessed in accordance with Eurocode, but for cast-in anchor bolts there exist only the European Assessment document EAD, which does not specified exactly the tested part or rebars. The study shown the difference between mechanical properties of bars and thread part. Following recommendations for manufacturer of anchor bolts. (see paper)

Round  2

Reviewer 1 Report

I have no other comments.

Author Response

Final paper after second review.

Reviewer 3 Report

See the attached document, which is the review for the Authors.

Author Response

Final paper after second roundd of reviews.

Content of experimental program was devided into more sections for better understanding. Methodology (methods of testing) was more explained. More references were added.
